# SWAP, SWITCH, and STABILIZE: Mechanisms of Kinetochore–Microtubule Error Correction

**DOI:** 10.3390/cells11091462

**Published:** 2022-04-26

**Authors:** Tomoyuki U. Tanaka, Tongli Zhang

**Affiliations:** 1Centre for Gene Regulation and Expression, School of Life Sciences, University of Dundee, Dundee DD1 5EH, UK; 2Department of Pharmacology and System Physiology, College of Medicine, University of Cincinnati, Cincinnati, OH 45219, USA; zhangtl@ucmail.uc.edu

**Keywords:** chromosome biorientation, kinetochore–microtubule interaction, error correction, initiation problem of biorientation (IPBO), Aurora B, chromosomal passenger complex (CPC), Mps1, Stu2, Dam1 complex (Dam1C), Ndc80 complex (Ndc80C)

## Abstract

For correct chromosome segregation in mitosis, eukaryotic cells must establish chromosome biorientation where sister kinetochores attach to microtubules extending from opposite spindle poles. To establish biorientation, any aberrant kinetochore–microtubule interactions must be resolved in the process called error correction. For resolution of the aberrant interactions in error correction, kinetochore–microtubule interactions must be exchanged until biorientation is formed (the SWAP process). At initiation of biorientation, the state of weak kinetochore–microtubule interactions should be converted to the state of stable interactions (the SWITCH process)—the conundrum of this conversion is called the initiation problem of biorientation. Once biorientation is established, tension is applied on kinetochore–microtubule interactions, which stabilizes the interactions (the STABILIZE process). Aurora B kinase plays central roles in promoting error correction, and Mps1 kinase and Stu2 microtubule polymerase also play important roles. In this article, we review mechanisms of error correction by considering the SWAP, SWITCH, and STABILIZE processes. We mainly focus on mechanisms found in budding yeast, where only one microtubule attaches to a single kinetochore at biorientation, making the error correction mechanisms relatively simpler.

## 1. Introduction

To maintain genetic integrity, eukaryotic cells must undergo accurate chromosome segregation in mitosis, prior to cell division. This process relies on correct interactions between kinetochores and microtubules (MTs) on the mitotic spindle. The kinetochore is a large protein complex that assembles at centromeric regions of chromosomes [1]. Meanwhile, the mitotic spindle is a bipolar structure consisting of MTs and associated proteins, in which the spindle poles are organized by the centrosomes in metazoan cells and by spindle pole bodies (SPBs) in yeast [2]. Spindle MTs dynamically switch between phases of growth and shrinkage, which are polymerization and depolymerization of tubulins, respectively [3].

Kinetochore–MT interactions take place in a stepwise manner [4] (Figure 1). Kinetochores initially attach to the lateral side of a MT (lateral attachment) extending from a spindle pole (spindle-pole MT) in a step evolutionarily conserved from yeast to vertebrate cells [5,6] (Figure 1, Step 2). The lateral attachment is often mediated by a short kinetochore-derived MT, which interacts with a spindle-pole MT (Step 1) [7,8,9,10]. Following this, the kinetochore on the MT lateral side is transported along a spindle-pole MT towards a spindle pole, which is driven by dynein in metazoan cells and by Kar3 (kinesin-14) in budding yeast [5,11,12] (Step 2). During this transport, the spindle-pole MT shrinks and its plus end catches up with the kinetochore, leading to the kinetochore being tethered at the MT end (end-on attachment) [13,14] (Step 3). As the end-on attached MT shrinks, the kinetochore is pulled towards a spindle pole. If sister kinetochores subsequently attach to MTs extending from the same pole, aberrant attachment is formed (Step 4) and this must be resolved through the process called error correction [15,16,17] (Step 5). Once sister kinetochores attach to MTs extending from opposite spindle poles (chromosome biorientation), kinetochore–MT interactions are stabilized due to tension applied across sister kinetochore and [15,16,17] (Step 6). To apply this tension, cohesion between sister chromatids is important; indeed, failure in forming sister chromatid cohesion causes extensive defects in chromosome biorientation [18,19]. Once biorientation is established for all chromosomes, spindle assembly checkpoint is satisfied and sister chromatid cohesion is removed, which triggers sister chromatid separation and segregation in anaphase [20,21] (Step 7).

In budding yeast *Saccharomyces cerevisiae*, kinetochores attach to the MTs extending from SPB(s) for the most part of the cell cycle [22]. However, during the S phase, kinetochores transiently detach from MTs, when kinetochores are disassembled due to centromere replication, and re-establish attachment soon after being reassembled [23]. Thus, budding yeast undergoes Steps 1–5 in Figure 1 during the S phase. Therefore, we propose that, in *S. cerevisiae*, S phase overlaps with prometaphase and there is no G2 or prophase [23]. In *S. cerevisiae*, only one MT attaches to a single kinetochore at biorientation (i.e., in metaphase), in contrast to most metazoan cells [22]. We expect that this configuration makes mechanisms of error correction relatively simpler.

Chromosome biorientation is crucial for correct chromosome segregation in anaphase (Figure 1, Steps 6 and 7) and error correction is fundamental for the establishment of biorientation (Steps 4 and 5). Meanwhile, earlier steps (Steps 1–3) are also important for the efficient establishment of chromosome biorientation. For example, kinetochore-derived MTs often promote efficient kinetochore interaction with the lateral side of a spindle-pole MT [10] (Step 1); the lateral side of a spindle-pole MT provides a larger surface for kinetochore interaction as compared with the MT end, thus, the lateral attachment is more suitable than the end-on attachment for the initial kinetochore interaction with a spindle-pole MT [5,6] (Step 2); the kinetochore transport along a spindle-pole MT indirectly facilitates the kinetochore interaction with another MT if the original interaction is lost in budding yeast [24] (Step 2). Moreover, in both yeast and metazoan cells, the end-on attachment is load bearing, i.e., withstands a larger force [25,26], and should be required to maintain chromosome biorientation, by which tension is applied across sister kinetochores. Therefore, the lateral attachment must be converted to the end-on attachment before or when biorientation is established (Steps 3 and 6).

In this review article, we discuss mechanisms of error correction to achieve chromosome biorientation (Figure 1, Steps 4–6). Cells undergo three important processes to achieve biorientation: First, aberrant kinetochore–MT attachments (Step 4) must be resolved through the exchange of kinetochore–MT interactions (Step 5) (SWAP). Second, the state of weak and unstable kinetochore–MT interactions should be switched to the state of robust and stable interactions [27] (SWITCH). Third, when biorientation is established, tension is applied across sister kinetochores, which stabilizes kinetochore–MT interactions (Step 6) (STABILIZE). We discuss these three processes, i.e., SWAP, SWITCH, and STABILIZE, mainly focusing on relevant mechanisms found in budding yeast. We also compare mechanisms in budding yeast and metazoan cells where relevant, but it is not our intention to comprehensively review error correction mechanisms in metazoan cells.

## 2. SWAP: How Kinetochore–MT Interactions Are Exchanged during Error Correction

### 2.1. Kinetochore–MT Interface Is Regulated by Aurora B Kinase and Other Factors for Error Correction

The kinetochore is a large protein complex consisting of dozens of components. While components of the inner kinetochore bind (or localize in the vicinity of) centromere DNA, components of the outer kinetochore form the kinetochore–MT interface. The Ndc80 complex (Ndc80C) and the Dam1 complex (Dam1C) are outer kinetochore components in budding yeast, which interact directly with MTs and play major roles in making the kinetochore–MT interface [26,28,29]. The Ndc80C is an integral part of the kinetochore, which consists of four proteins (Ndc80, Nuf2, Spc24, and Spc25), and is essential for both lateral and end-on attachment [5]. On the other hand, during the lateral attachment, the Dam1C (consisting of ten different proteins including Dam1) is not a part of the kinetochore but is present at the dynamic MT plus end [14,23]. When the end-on attachment is established following the lateral attachment (Figure 1, Step 3), Ndc80Cs (already at the kinetochore) interact with Dam1Cs localizing at the MT end [30,31,32,33]. This Ndc80C–Dam1C interaction stabilizes the end-on kinetochore–MT interface and withstands tension across sister kinetochores when biorientation is established (Figure 1, Step 6). It is suggested that three unstructured regions of Dam1C components (the C-termini of Dam1, Ask1, and Spc34) interact with three different regions of Ndc80/Nuf2 (components of the Ndc80C) [34,35,36].

As mentioned in Section 1, aberrant kinetochore–MT interactions are often formed in early mitosis (Figure 1, Step 4). More specifically, syntelic attachments are often formed, in which both sister kinetochores interact with MTs extending from the same spindle pole. Syntelic attachments must be resolved through the error correction process (Figure 1, Step 5). For this, Aurora B kinase (called Ipl1 in budding yeast) plays a central role from yeast to humans [4,37,38,39,40,41]. Aurora B forms the chromosomal passenger complex (CPC) with INCENP, Survivin, and Borealin (called Sli15, Bir1, and Nbl1, respectively, in budding yeast) [42]. The CPC is recruited to the centromere and inner kinetochore [43,44,45]. When tension is not applied across sister kinetochores (e.g., during syntelic kinetochore–MT attachment), the Dam1C and Ndc80C are phosphorylated by Aurora B, which weakens and subsequently disrupts kinetochore–MT interactions, in budding yeast [46,47,48]. Phosphorylation of the Dam1C by Aurora B is essential for error correction, while Ndc80C phosphorylation (at the N-terminus of Ndc80) modestly contributes to error correction. Crucially, the three unstructured regions of Dam1C components (the C-termini of Dam1, Ask1, and Spc34), which are involved in interactions with Ndc80/Nuf2 [34,35], are important targets of Aurora B for error correction [46]. The evidence suggests that, when tension is low, Aurora B phosphorylates these regions, causing disruption of the Ndc80C–Dam1C interaction and loss of the end-on attachment [33,35,36,49].

In addition to Aurora B kinase, other factors also regulate kinetochore–MT interactions for error correction. For example, Mps1 kinase plays an important role in resolving aberrant kinetochore–MT interactions and this function is conserved from yeast to human cells [50,51,52,53,54]. Mps1 kinase is recruited to the kinetochore through the interaction with the Ndc80C [55,56,57] and phosphorylates outer kinetochore components Ndc80, Spc105, and Ska3 to promote chromosome biorientation [58,59,60]. Moreover, although Stu2 (and its orthologue ch-TOG in humans) operates as a MT polymerase, a fraction of Stu2 is recruited to the kinetochore by Ndc80C [61,62]. It is suggested that this fraction of Stu2 destabilizes kinetochore–MT interactions when tension is low, independently of MT dynamics and Aurora B kinase [61,63].

### 2.2. Aurora B Differentially Regulates Kinetochore Interaction with the Side and End of a MT

For the efficient establishment of chromosome biorientation, aberrant kinetochore–MT interactions should be rapidly removed. In addition, new interactions need to be frequently formed. However, since the action of Aurora B disrupts kinetochore–MT interactions when tension is low, a straightforward assumption would be that Aurora B also discourages new kinetochore–MT interactions. Alternatively, new interactions may be formed, if targets of Aurora B are rapidly dephosphorylated at kinetochores after disruption of aberrant kinetochore–MT interaction but before the formation of new interaction. However, such a mechanism is not identified or implied. Then, how does Aurora B still allow the formation of new kinetochore–MT interactions?

A clue to this question was found when it was studied how Aurora B regulates kinetochore interaction with the side and plus end of a MT (lateral and end-on attachment, respectively) in early mitosis of budding yeast. Phospho-mimic mutants at the Dam1 C-terminus (at Aurora B phosphorylation sites) weakened the end-on attachment but did not affect the lateral attachment [33]. In addition, phospho-mimic mutants at the Ndc80 N-terminus (at Aurora B phosphorylation sites) modestly weakened the end-on attachment but did not affect the lateral attachment. Thus, it was suggested that the end-on and lateral attachment is differentially regulated by Aurora B kinase [33].

Based on these results, it is proposed that this differential regulation drives the exchange of kinetochore–MT interactions during error correction [33], as follows: An end-on attachment (forming an aberrant kinetochore–MT interaction) is disrupted, mainly due to Dam1 phosphorylation by Aurora B (Figure 2A, Steps 1 and 2). Subsequently, the lateral attachment is formed on another MT, since this is not inhibited by Aurora B-dependent Dam1 phosphorylation (Steps 3 and 4). Then, the lateral attachment is converted to the end-on attachment on the same MT and, if this leads to aberrant attachment, it must be resolved again by Aurora B (Step 1). However, if biorientation is formed (Step 5), tension is subsequently applied across sister kinetochores, which stabilizes the kinetochore–MT interactions (Step 6).

More recently, using in vitro reconstitution of a kinetochore–MT interface, the relative strengths of the lateral and end-on attachments were directly compared [49]. Although the end-on attachment was stronger with non-phosphorylated Dam1, the lateral attachment became stronger with phospho-mimic Dam1 mutants (at Aurora B phosphorylation sites) [49]. Thus, Aurora B-dependent Dam1 phosphorylation switches the relative strength of the two modes of kinetochore–MT attachment. It is likely that this switch promotes the exchange of kinetochore–MT interactions, i.e., from the end-on attachment on one MT to the lateral attachment on another MT (Figure 2A, Steps 1–4).

### 2.3. Loss of the End-on Kinetochore–MT Attachment and Resolution of a Syntelic Attachment

Although Aurora B-dependent phosphorylation of Dam1C and Ndc80C weakens the end-on attachment, several pieces of evidence imply that this is insufficient for efficient disruption of the end-on attachment, as follows: First, while the kinetochore was transported by depolymerization of an end-on attached MT in yeast cells, the end-on attachment was rarely lost [14]. Second, phospho-mimic Dam1 mutants (at Aurora B phosphorylation sites) led to only slow (over 30 min or longer) detachment from the MT end in cells [33]. Third, phospho-mimic Dam1 mutants caused rare detachment of a purified kinetochore particle from the MT end in vitro [49]. Nonetheless, to resolve a syntelic attachment, the end-on attachment must be disrupted at least at one of two sister kinetochores (Figure 2A, Step 1 to 2). How is this achieved?

For this, we propose the following mechanism: When a syntelic attachment is formed, two MTs from the same spindle pole form the end-on attachment to sister kinetochores (Figure 2A, Step 1 and Figure 2B). In such a situation, even a slight difference in their dynamics (e.g., rate of MT growth or shrinkage, timing of rescue and catastrophe) would generate twisting forces on sister kinetochores in a syntelic attachment (Figure 2Bb,d and e) [15]. The twisting forces would cause disruption of already weakened end-on attachment (due to Aurora B-dependent kinetochore phosphorylation) to one sister kinetochore. This disruption may happen especially at kinetochores that are “pushed” away from a spindle pole by a polymerizing MT (Figure 2B, kinetochores marked with #), due to structural constraints of kinetochore components such as Ndc80C (which might be forced to bend sharply in this situation). Once the end-on MT attachment is disrupted at one sister kinetochore, the twisting force should be released, allowing the end-on attachment to the other sister kinetochore to remain, even if it is weakened (Figure 2A, Step 2). Such a mechanism would have an advantage, i.e., it would avoid a simultaneous loss of MT attachments at both sister kinetochores. This should contribute to preventing a chromosome (a pair of sister chromatids) from drifting away from the spindle during error correction. If a chromosome drifts away, it must be caught again on a MT extending from a spindle pole, but this is time-consuming [10,64] and would delay the establishment of biorientation.

## 3. SWITCH: How a Low-Tension State Is Converted to a High-Tension State at Initiation of Biorientation

### 3.1. Initiation Problem of Biorientation (IPBO): Transition from Low-to High-Tension State

As discussed in Section 2.1, a kinetochore–MT interaction is unstable when tension is not applied (low-tension state) and becomes stable only after tension is applied across sister KTs (high-tension state) [4,65]. The transition from a low-tension to high-tension state is a fundamental step in establishing stable biorientation. However, it is still unclear how this transition becomes feasible because, paradoxically, while stable attachment requires high tension, high tension can only be applied when a stable attachment is present (Figure 3A). Thus, it seems difficult for a kinetochore–MT interaction to acquire high tension or stable attachment when neither is present [27]. In other words, because the low-tension state is stable, any minor displacement from this state would be reverted, and the transition from a low- to high-tension state appears difficult (Figure 3A). For example, immediately after biorientation is formed (Figure 2A, Step 5), the kinetochore–MT interaction is still weak as tension is still low. Therefore, the interaction would be lost when some tension is applied (between Steps 5 and 6 in Figure 2A), before a full tension is able to stabilize the kinetochore–MT interaction (Figure 2A, Step 6). The conundrum regarding the transition from low- to high-tension state is called the “initiation problem of biorientation (IPBO)” [27].

The IPBO was defined more precisely using a mathematical model, based on the influence diagram [27] (Figure 3B). Tension suppresses the ability of Aurora B kinase to weaken the kinetochore–MT interactions at each sister kinetochore. Then, robust kinetochore–MT interactions at both sister kinetochores promote tension in these interactions. The influence diagram gives rise to a bistable system, in which a low-tension state and a high-tension state are represented by white and black dots, respectively, in Figure 3C. For the transition from a low-tension state to a high-tension state (each positioned at the “bottom of basin”), a separatrix (“watershed” between the “two basins” on the bottom left and the top right) must be traversed [27] (Figure 3C). However, this would be a difficult task without a mechanism facilitating this transition. A computer model of a kinetochore–MT interaction in fission yeast illustrated that the IPBO, i.e., difficulty in transition from low- to high-tension state, is a general challenge in mitosis [66].

### 3.2. Possible Solutions for the Initiation Problem of Biorientation (IPBO)

To establish chromosome biorientation, a low-tension state of kinetochore–MT interactions must be converted to a high-tension state. As it is difficult for this transition to occur spontaneously, we require a mechanism to facilitate this transition. For this, the following mechanisms are proposed:

First, Tubman et al. suggested that phosphorylation by Aurora B kinase at multiple kinetochore sites plays an important role [67]. This model is based on the hypothesis that low tension usually leads to a medium number of phosphorylations, but occasionally results in a critical high number of phosphorylations. While the critically high phosphorylation causes kinetochore detachment from a MT, the medium phosphorylation allows time for transition to a high-tension state as it does not disrupt kinetochore–MT attachment when biorientation is initiated (i.e., immediately after the other sister kinetochore interacts with a MT from the opposite spindle pole, Figure 2A, Step 5) [67].

Second, the conversion from the lateral to the end-on kinetochore–MT attachment may play a key role [33]. As discussed in Section 2.1, the Dam1C is not a part of the kinetochore during the lateral attachment. When the lateral attachment is converted to the end-on attachment, the Dam1C associates with Ndc80C to form the end-on kinetochore configuration (see Figure 4, top). The CPC (containing Aurora B) at centromeres and inner kinetochores would be able to phosphorylate Dam1C only after the end-on attachment is formed. However, as physical adaptation often precedes chemical adaptation [68], it may take some time to fully phosphorylate Dam1C (leading to kinetochore detachment from a MT) after the establishment of end-on attachment. This would make a time window allowing the transition from low- to high-tension state when biorientation is initiated (Figure 2A, Step 5) (Zhang, Novak, Tanaka et al., unpublished).

Third, the end-on kinetochore–MT attachment is load-bearing, i.e., withstands a large force [25,26]. While it was thought that the lateral kinetochore–MT attachment may not be load-bearing, it was recently shown that the lateral attachment can withstand a force to some extent in metazoan cells [69]. Given this, if a sister kinetochore is caught on a MT while the other sister is on the lateral side of another MT extending from the opposite pole, the lateral attachment may withstand a modest force during the transition from low- to high-tension state. Then, when the end-on attachment is formed on both sister kinetochores, biorientation may be established with tension fully applied (Figure 2A, Step 6).

Fourth, the solution for the IPBO may come from the evidence that Aurora B-dependent phosphorylation of Dam1C (and Ndc80C) is insufficient for disruption of the end-on attachment (see Section 2.3). The end-on attachment may be efficiently disrupted only when both (1) phosphorylation of Dam1C (and Ndc80C) makes the kinetochore–MT interaction weak with low tension and (2) the syntelic attachment applies twisting forces between sister kinetochores (Figure 2B). If so, it would allow transition from a low- to high-tension state as the kinetochore–MT interaction is not disrupted by a pulling force following initiation of biorientation (since no twisting force is applied, Figure 2A, Steps 5 to 6).

Although all these models are based on biological evidence, each model has its own limitation as follows: The first model may make error correction relatively inefficient because kinetochore detachment from a MT would occur only with moderate frequency when critically high phosphorylation happens with low tension. With the second model, the time window enabling the transition from low-to high-tension state may be narrow if phosphorylation of Dam1C is delayed only modestly following the end-on attachment establishment. With the third model, since the lateral attachment is rapidly converted to the end-on attachment [13,23], the time window enabling the transition may be narrow. The third mechanism may work in metazoan cells, but not in budding yeast, because the lateral attachment is converted to the end-on attachment before two spindle poles separate (which is followed by error correction and biorientation) [23]. In the fourth model, when the kinetochore–MT interaction is weakened by the Aurora B activity, it should have a specific property, i.e., it should be easily disrupted by a twisting force due to syntelic attachment (Figure 2A, from Step 1 to 2 and Figure 2B), but not by a pulling force following initiation of biorientation (Figure 2A, from Step 5 to 6). It is plausible that multiple mechanisms redundantly work to allow the transition from low- to high-tension state. In this way, these mechanisms may compensate for each other’s limitations to ensure the transition without failure.

Meanwhile, there may be more mechanisms to facilitate the transition from low- to high-tension state. In principle, any mechanism that would delay kinetochore detachment from a MT while tension is low could facilitate this transition. Nevertheless, the delay cannot reduce the detachment rate too much, otherwise the resolution of erroneous kinetochore–MT interactions would become inefficient.

## 4. STABILIZE: How the Kinetochore–MT Interaction Is Stabilized When Biorientation Is Established

### 4.1. Aurora B Localization Sites at Centromeres/Kinetochores and the Aurora B Spatial Separation Model

To resolve an aberrant kinetochore–MT attachment, kinetochore–MT interactions are exchanged in the process of error correction (Figure 1, Steps 4 and 5). However, when biorientation occurs, a low-tension state is converted to a high-tension state (see Section 3) and tension is applied across sister kinetochores, stabilizing the kinetochore–MT interaction with biorientation (Figure 1, Step 6). It is widely accepted that tension applied to the kinetochore–MT interaction regulates the stability of this interaction [16,70]. However, it is still elusive how tension actually stabilizes kinetochore–MT interactions.

As discussed in Section 2.1, Aurora B kinase plays a central role in weakening kinetochore–MT interactions when tension is low. However, when biorientation is established and high tension is applied, this function of Aurora B must cease or be overcome to avoid further exchange of kinetochore–MT interactions. To consider mechanisms for this, we should first understand where Aurora B localizes to promote error correction. Several studies have reported that the CPC (consisting of Aurora B, INCENP, Survivin, and Borealin [42]) is recruited to the centromere. This recruitment relies on (1) histone H3 phosphorylation by Haspin kinase and (2) histone H2A phosphorylation by Bub1 kinase [71,72,73,74]. The H3 phosphorylation recruits Survivin, while the H2A phosphorylation recruits Shugoshin that in turn binds Survivin or Borealin [71,72,73,74,75]. Because Bub1 is recruited to kinetochores and Haspin is enriched by association with cohesin, the combined effect of (1) and (2) is highest at centromeres, where CPC is enriched during prometaphase and metaphase [74].

In budding yeast, Survivin is required to recruit the CPC to centromeres. However, it has been shown that Survivin can become dispensable for biorientation in this organism [76]. If error correction requires CPC localization at centromeres (or somewhere close to it), there must be a Survivin-independent CPC recruitment mechanism. Indeed, it was found that, independently of Survivin, INCENP directly interacts with the Mcm21–Ctf19 subcomplex (CENP-O–CENP-P subcomplex in metazoan cells) at the inner kinetochore for recruitment of Aurora B, in budding yeast [43,44]. The two mechanisms for CPC recruitment, i.e., Survivin-dependent centromere recruitment and Survivin-independent inner kinetochore recruitment, work redundantly to promote chromosome biorientation, though the Survivin-dependent mechanism is usually predominant [43,44]. If both recruitment mechanisms are defective, most sister kinetochore pairs fail to establish biorientation.

Since CPC localizes at the centromere and inner kinetochore, the following model explains how kinetochore–MT interactions are stabilized in a tension-dependent manner in budding yeast: When tension is applied, the kinetochore is stretched, which spatially separates Aurora B from its outer kinetochore substrates (Ndc80 N-terminus and Dam1C components) whose phosphorylation is crucial for error correction (see Section 2.1) (Figure 4). The long coiled-coil region of the Ndc80C exceeds INCENP in length when stretched, thus, contributing to this spatial separation [77,78] (Figure 4, bottom). The spatial separation leads to dephosphorylation of the outer kinetochore substrates, which stabilizes the kinetochore attachment to the MT end; this model is called Aurora B spatial separation model [41,79]. On the other hand, a kink in the middle of Ndc80C allows the Ndc80C to bend flexibly if tension is low [80,81,82], which enables Aurora B to access its outer kinetochore substrates with low tension (Figure 4, top).

A series of evidence supports the Aurora B spatial separation model in budding yeast and mammalian cells, as follows: First, phosphorylation of outer kinetochore components by Aurora B is reduced when tension is high [48,79,83]. Second, ectopic targeting of Aurora B/INCENP to the outer kinetochore destabilizes kinetochore–MT interactions during metaphase in human cells [84] and in budding yeast (Garcia-Rodriguez and Tanaka, unpublished). Crucially, the Aurora B spatial separation model explains why CPC relocalizes from the centromere/inner kinetochore to the spindle mid-zone at the anaphase onset [42]. If Aurora B were to remain at the centromere/inner kinetochore in anaphase, kinetochore–MT interactions would be destabilized since the loss of cohesion reduces tension in this interaction at anaphase onset. In fact, this was demonstrated by experiments using INCENP (and other) mutants, with which Aurora B remained at kinetochores during anaphase [85,86,87,88].

### 4.2. Other Mechanisms Regulating the Kinetochore–MT Interactions in a Tension-Dependent Manner

A series of evidence supports the Aurora B spatial separation model, as discussed in Section 4.1. However, it is plausible that other mechanisms are additionally required to stabilize kinetochore–MT interactions when tension is applied as a result of biorientation establishment. Such mechanisms may involve dynamic regulation of CPC (including Aurora B). For example, the following mechanisms are suggested: First, CPC may be released from its localization sites and reach the substrates of Aurora B kinase at the outer kinetochores [65,89]. If the localization sites are closer to the substrates, CPC would reach substrates more frequently. Therefore, this is consistent with the Aurora B separation model, but it facilitates Aurora B to reach its substrates further away. Second, CPC components may be stretched by tension and this conformational change may reduce the Aurora B activity. For example, it is proposed that INCENP has a spring-like property to regulate the Aurora B activity in a tension-dependent manner [78,90]. Third, a small fraction of CPC may be present at the outer kinetochore, near the kinetochore–MT interface, in early mitosis [45,91]. It is suggested that the localization of CPC at the outer kinetochore is reduced when tension is applied due to biorientation, which stabilizes a kinetochore–MT interaction [92]. Fourth, INCENP and Borealin (both are CPC components) have MT-binding domains [78,93,94,95]. It is proposed that phosphorylation of Aurora B substrates at the outer kinetochore is regulated by MT binding of the CPC in a tension-dependent manner [96]. Fifth, Borealin undergoes phase separation at least in vitro, and it is proposed that, when aberrant kinetochore–MT interactions are formed, this phase separation facilitates the CPC accumulation at centromeres/kinetochores [97].

Alternatively, kinetochore–MT interactions may be regulated in a tension-dependent manner by factors other than Aurora B, in the following ways: First, when tension is applied, the rate of MT catastrophe (conversion from MT polymerization to depolymerization) is reduced in vitro, independently of Aurora B. In turn, this decreases the rate of kinetochore detachment from a MT [98]. Second, as discussed in Section 2.1, Mps1 kinase promotes the resolution of aberrant kinetochore–MT interactions. The Mps1 activity at kinetochores is enhanced in the absence of MT attachment but reduced when biorientation is established [55,56,99], thus Mps1 may be involved in a tension-dependent regulation of kinetochore–MT interactions. Third, the fraction of Stu2 localizing at the kinetochore may have dual functions in stabilizing and destabilizing kinetochore–MT interactions when high and low tension is applied, respectively [61,63,100]. These functions of Stu2 appear to be independent of Aurora B kinase and MT dynamics [63]. Fourth, PP1 and PP2A phosphatases are recruited to the kinetochore by KNL1 (Spc105 in budding yeast) and other factors to regulate the spindle-assembly checkpoint [101,102,103,104,105]. It is also suggested that PP1 and PP2A regulate kinetochore–MT interactions more directly [106,107,108,109,110]. In budding yeast, Glc7, a PP1 orthologue, plays a major role in counteracting the Aurora B kinase activity to regulate kinetochore–MT interactions [111,112,113]. The regulation of kinetochore–MT interactions by phosphatases may be tension dependent; for example, the conformational change of Aurora B substrates, which is caused by tension, may help phosphatases to dephosphorylate them.

## 5. Concluding Remarks

In this article, we have reviewed mechanisms regulating error correction of kinetochore–MT interactions, mainly focusing on mechanisms in budding yeast. The use of budding yeast *S. cerevisiae* as a model organism has greatly contributed to research on error correction because (1) versatile genetics in this organism help to identify important regulators for error correction; (2) only one MT attaches to a single kinetochore at biorientation in budding yeast [22], which makes the study of error correction simpler; and (3) many mechanisms promoting error correction are essentially conserved from yeast to humans. Over the last 20 years, components of the kinetochore–MT interface, such as Ndc80C and Dam1C, have been found and studied in detail. Moreover, several regulators for error correction, such as Aurora B, Mps1, Stu2, and phosphatases, have been identified and characterized. However, it is still not completely understood how kinetochore–MT interactions are exchanged during error correction, how a low-tension state is converted to a high-tension state when chromosome biorientation is initiated, and how kinetochore–MT interactions are stabilized in a tension-dependent manner when biorientation is established. The combined efforts in genetics, cell biology, biochemistry, structural analyses, biophysics, and mathematical modeling will advance research in this important field. Chromosome biorientation is at the heart of mechanisms ensuring correct chromosome segregation and genetic integrity in proliferating cells. Failure in establishing biorientation leads to human diseases characterized by chromosome instability and aneuploidy [114,115,116]. To elucidate how these diseases develop, it is essential to understand mechanisms of error correction leading to chromosome biorientation.

## Figures and Tables

**Figure 1 cells-11-01462-f001:**
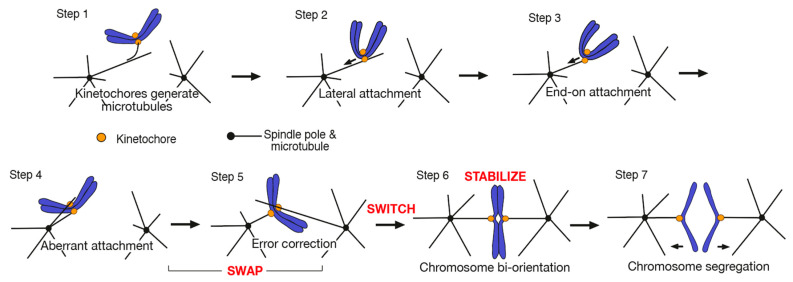
Diagram shows kinetochore–MT interactions during mitosis. Steps 1–6 show the processes, by which correct kinetochore–MT interactions are established, prior to chromosome segregation [4]. Step 7 shows chromosome segregation in anaphase. Each step is explained in the text. SWAP, SWITCH, and STABILIZE are the processes of error correction and the main topics of this review article.

**Figure 2 cells-11-01462-f002:**
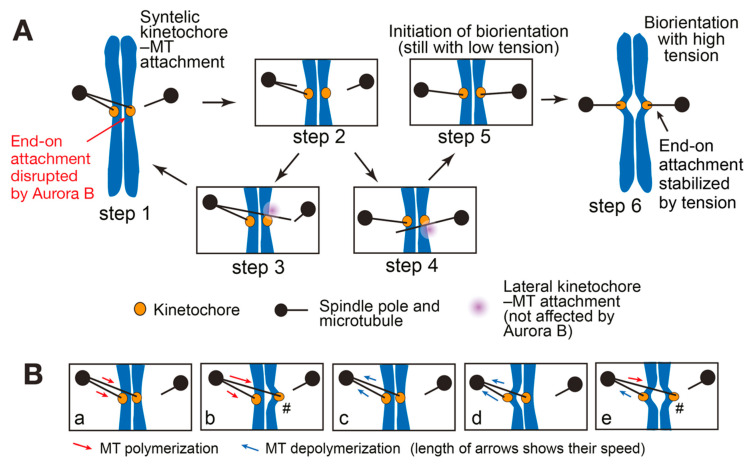
(**A**) Diagram shows a model of the error correction process, in which an aberrant kinetochore–MT interaction (syntelic attachment) is resolved to establish biorientation [33]. Each step is explained in the text. This figure was taken from Doodhi et al. [49] after modification; (**B**) the dynamics of kinetochore-attached MTs and predicted effects on sister kinetochores in a syntelic attachment; in (**a**,**c**) the motion of two sister kinetochores is coordinated; in (**b**,**d**,**e**), different dynamics of kinetochore-attached MTs generate twisting forces on sister kinetochores. The kinetochore–MT interaction may be disrupted especially at kinetochores marked with # (see text). Note that the twisted kinetochores in (**b**,**d**,**e**) do not represent an application of full tension (in contract to Step 6 in (**A**)) since an end-on attachment would be disrupted at one of the sister kinetochores before full tension is applied.

**Figure 3 cells-11-01462-f003:**
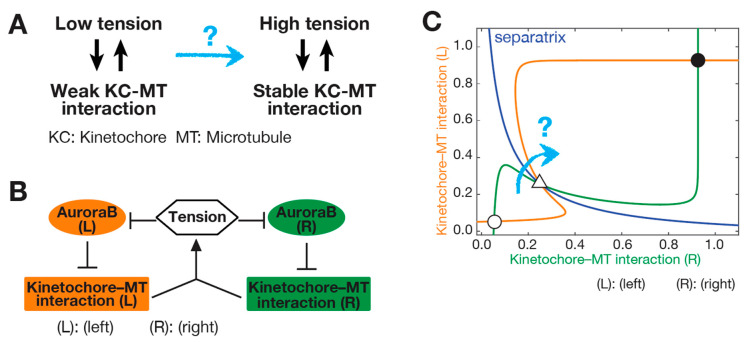
(**A**) Diagram explains the transition from a low- to high-tension state when chromosome biorientation is initiated; the difficulty in this transition is highlighted by “?”; (**B**) influence diagram shows how tension, Aurora B kinase, and kinetochore–MT interaction affect each other, at the left (orange) and right (green) sister kinetochores [27]; (**C**) diagram shows two stable states (low-tension state (white circle), high-tension state (black circle)), and the separatrix to be crossed (blue). The low-tension state (white circle) is illustrated in Figure 2A, Step 5, while the high-tension state (black circle) is shown in Figure 2A, Step 6. The white triangle indicates the unstable “saddle” point [27]. The bistable system in this diagram emerges from the influence diagram in B and illustrates the difficulty (highlighted by “?”) in transition from the low- to high-tension state (initiation problem of biorientation, IPBO) [27].

**Figure 4 cells-11-01462-f004:**
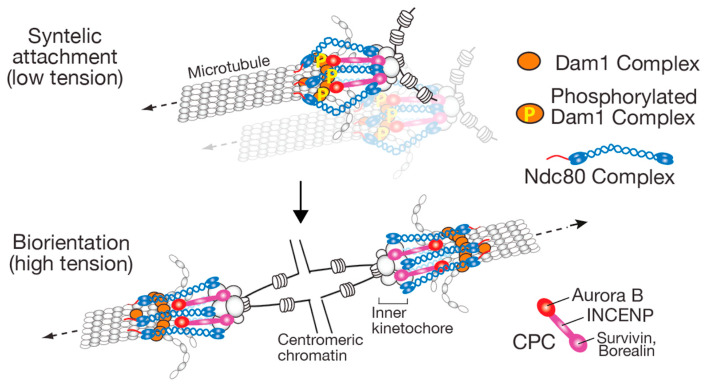
Diagram represents Aurora B spatial separation model, which explains how tension, applied across sister kinetochores, stabilizes kinetochore–MT interactions [4]. CPC is recruited to the inner kinetochore and centromeric nucleosome. During syntelic attachment (aberrant kinetochore–MT interactions), Aurora B reaches Dam1C and phosphorylates its components, which weakens and disrupts kinetochore attachment to the MT end (**top**). When biorientation is established, Ndc80Cs are stretched and Aurora B cannot reach Dam1C, whose dephosphorylation stabilizes kinetochore–MT interactions (**bottom**). Although CPCs are shown only at the inner kinetochore in the diagram for simplicity, CPCs at the centromeric nucleosome (not shown here) also show similar behaviors. The figure was taken from [17] after modification. The CPC consists of Aurora B, INCENP, Survivin, and Borealin, which are called Ipl1, Sli15, Bir1, and Nbl1, respectively, in budding yeast [42].

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
