# Peer review of "SWAP, SWITCH, and STABILIZE: Mechanisms of Kinetochore–Microtubule Error Correction"

_cells, 2022, doi:10.3390/cells11091462_

Round 1
Reviewer 1 Report
To accomplish faithful chromosome segregation, a mitotic cell needs to establish chromosome biorientation, in which all pairs of sister kinetochores attach to opposite spindle poles through spindle microtubules. Because the initial spindle-chromosome capture is typically error-prone, a dividing cell relies on error correction machineries to resolve such erroneous attachments before reaching biorientation. The authors dissect the error correction process with three sequential steps: 1) Swap, in which a kinetochore can exchange its weakly bound microtubule between two spindle poles; 2) Switch, in which a weakly bound kinetochore transits to some strong binding states; and 3) Stabilize, in which a bioriented sister kinetochore pair sets up tension to strength kinetochore-microtubule binding. The article is insightful and well written and should be published. I have a few small suggestions to clarify the ideas:
- Line 197. It’s unclear if the “shearing force” is different from “tension” discussed elsewhere in the manuscript, and thus including a cartoon figure can help guide readers to understand the concept.
- Starting from Line 208. IPBO may be simplified as a kinetic race problem. To allow biorientation, initial attachment cannot be released rapidly (slowed off-rate), and the attachment of the unbound kinetochore needs to be fast enough (fast on-rate). By dissecting the IPBO this way, the authors may also discuss other possible mechanisms to help overcome IPBO, such as (but not limited to):
- Lateral attachment generates tension and thereby prohibiting release (PMID 28536121 in mammalian, and papers published by the authors such as PMID 25751138).
- Unbound kinetochores can expand to promote search-and-capture in mammalian cells (PMID 29915359).
- High spindle microtubule density can also promote attachment and thereby resolving the IPBO.
Author Response
Line 197. It’s unclear if the “shearing force” is different from “tension” discussed elsewhere in the manuscript, and thus including a cartoon figure can help guide readers to understand the concept.
> Following this suggestion, we have added new Figure 2B to explain how twisting forces are generated by different kinetics of kinetochore-attached MTs during syntelic attachment. Note that we have replaced the word ‘shearing forces’ with ‘twisting forces’ to highlight the kinetochore deformation shown in Figure 2B. Moreover, in Section 2.3, we have explained how twisting forces could cause disruption of kinetochore–MT interaction. Before a full tension is applied, the end-on attachment would be disrupted at one of sister kinetochores in syntelic attachment, as mentioned in the Figure 2B legend.
Starting from Line 208. IPBO may be simplified as a kinetic race problem. To allow biorientation, initial attachment cannot be released rapidly (slowed off-rate), and the attachment of the unbound kinetochore needs to be fast enough (fast on-rate). By dissecting the IPBO this way, the authors may also discuss other possible mechanisms to help overcome IPBO, such as (but not limited to):
- Lateral attachment generates tension and thereby prohibiting release (PMID 28536121 in mammalian, and papers published by the authors such as PMID 25751138).
- Unbound kinetochores can expand to promote search-and-capture in mammalian cells (PMID 29915359).
- High spindle microtubule density can also promote attachment and thereby resolving the IPBO.
> The first mechanism above would achieve ‘slowed off-rate’ and may indeed facilitate the transition from low- to high-tension state. We have added this mechanism to Section 3.2. The second and third mechanisms above would achieve ‘fast on-rate’. However, many mechanisms facilitate ‘fast on-rate’ to generally promote kinetochore–MT interaction, rather than to specifically promote the transition from low- to high-tension state. In Section 3.2, we would rather focus on mechanisms achieving ‘slowed off-rate’ because they are more specific mechanisms facilitating the transition. Nonetheless, in line with the reviewer’s suggestion, we have added the following statement in Section 3.2: ‘there may be more mechanisms to facilitate the transition from low- to high-tension state. In principle, any mechanism, which would delay kinetochore detachment from a MT while tension is low, could facilitate this transition’.
Reviewer 2 Report
This manuscript summarizes the recent advance in the understanding of the molecular mechanisms that ensure chromosome bipolar attachment with focus on budding yeast. This topic is interesting because of the critical role of correct kinetochore-microtubule connection in genome integrity. The authors have done an excellent job to present possible mechanisms of kinetochore-microtubule interaction, error correction, and stabilization. Some minor issues need to be addressed before publication.
-The CPC plays a key role in the error correction of kinetochore-microtubule attachment. It would be better to include a diagram showing the components of CPC in budding yeast and mammalian cells.
-One major function of the CPC is the correction of tensionless syntelic attachment in budding yeast. Therefore, it is necessary to introduce the conditions that increase the frequency of syntelic attachment in budding yeast.
-The potential mechanisms that regulates CPC activity have been well explained in this manuscript. The regulation of PPase activity at the kinetochore also changes the status of CPC-medicated protein phosphorylation. It would be better to introduce more about the regulation of PPase activity at the kinetochore.
Author Response
-The CPC plays a key role in the error correction of kinetochore-microtubule attachment. It would be better to include a diagram showing the components of CPC in budding yeast and mammalian cells.
> The components of the CPC and their relative positions within the CPC are shown at the bottom right corner of Figure 4. Following the above suggestion, we have added the names of CPC components in budding yeast to the Figure 4 legend in the revised manuscript.
-One major function of the CPC is the correction of tensionless syntelic attachment in budding yeast. Therefore, it is necessary to introduce the conditions that increase the frequency of syntelic attachment in budding yeast.
> It is difficult to directly visualize syntelic attachment in budding yeast, although various data suggest it is present in this organism (e.g. Dewar et al 2004 from Tanaka group). Therefore, it is difficult to define the conditions, which increase the frequency of syntelic attachment in budding yeast.
-The potential mechanisms that regulates CPC activity have been well explained in this manuscript. The regulation of PPase activity at the kinetochore also changes the status of CPC-medicated protein phosphorylation. It would be better to introduce more about the regulation of PPase activity at the kinetochore.
> It is not very well established how PPase activity is regulated to change kinetochore–MT interaction in budding yeast, apart from the recruitment mechanisms of PPase to the kinetochore (which are already discussed in the manuscript). Nonetheless, following this suggestion from the reviewer, we have explained more about PPase in the revised manuscript by stating that, in budding yeast, Glc7, a PP1 orthologue, plays a major role in counteracting the Aurora B kinase activity to regulate kinetochore–MT interaction.
Reviewer 3 Report
This review describes our knowledge of the mechanisms involved in chromosome biorientation during prometaphase to reach metaphase. The authors describe how microtubules attach to kinetochores and how incorrect attachments are corrected to result in biorientation of each chromosome and stabilization under tension. The central role of Aurora B is described.
I enjoyed reading this review, it is well written and illustrated.
I have only very minor comments
Line 109-110
Use “Dam1C” instead of “Dam1 complex”
Line 115-116
“It is suggested that three unstructured regions of Dam1C components (Dam1, Ask1, Spc34/Spc19) interact with three different regions of the Ndc80C [34,35].”
is it three different regions in three different components of Ndc80C? or three different regions of the same component of Ndc80C?
Line 208
- SWITCH: How a low-tension state is converted to a high-tension state at initiation of biorientation
3.2. Possible solutions for the initiation problem of biorientation (IPBO)
This paragraph is very interesting.
The mechanisms involved to convert low-tension state to a high-tension state are difficult to understand and therefore to imagine.
Can we imagine a stochastic attachment-detachment mechanism that when it starts to become bioriented induces a progressive distancing of Aurora B from its substrates Dam1C and then a progressive stabilization of the biorientation?
Author Response
Line 109-110. Use “Dam1C” instead of “Dam1 complex”
> Thank you for indicating this error. It has been amended in the revised manuscript.
Line 115-116. “It is suggested that three unstructured regions of Dam1C components (Dam1, Ask1, Spc34/Spc19) interact with three different regions of the Ndc80C [34,35].” Is it three different regions in three different components of Ndc80C? or three different regions of the same component of Ndc80C?
> It is more precise to state that three unstructured regions of Dam1C components (the C-termini of Dam1, Ask1 and Spc34) interact with three different regions of Ndc80/Nuf2 (components of the Ndc80C). This has been amended accordingly, in the revised manuscript.
Line 208. SWITCH: How a low-tension state is converted to a high-tension state at initiation of biorientation – Possible solutions for the initiation problem of biorientation (IPBO). This paragraph is very interesting. The mechanisms involved to convert low-tension state to a high-tension state are difficult to understand and therefore to imagine. Can we imagine a stochastic attachment-detachment mechanism that when it starts to become bioriented induces a progressive distancing of Aurora B from its substrates Dam1C and then a progressive stabilization of the biorientation?
> Progressive distancing and stabilization alone would not be sufficient for the transition from low- to high-tension state. However, in principle, any mechanism, which would delay kinetochore detachment from a MT while tension is low, could facilitate this transition. We have explained this in Section 3.2 of the revised manuscript.